# Apoptosis-Inducing Potential of Selected Bromophenolic Flame Retardants 2,4,6-Tribromophenol and Pentabromophenol in Human Peripheral Blood Mononuclear Cells

**DOI:** 10.3390/molecules27165056

**Published:** 2022-08-09

**Authors:** Anna Barańska, Paulina Sicińska, Jaromir Michałowicz

**Affiliations:** Department of Biophysics of Environmental Pollution, Faculty of Biology and Environmental Protection, University of Lodz, Pomorska Str. 141/143, 90-236 Lodz, Poland

**Keywords:** 2,4,6-tribromophenol, pentabromophenol, peripheral blood mononuclear cells, apoptosis, cytosolic calcium ion level, transmembrane mitochondrial potential, caspase activation, PARP-1 cleavage, chromatin condensation

## Abstract

(1) Background: 2,4,6-Tribromophenol (2,4,6-TBP) and pentabromophenol (PBP) are utilized as brominated flame retardants (BFRs) in order to reduce the combustion of materials used in various utility products. The presence of 2,4,6-TBP and PBP has been reported in environmental samples as well as in inhaled air, dust, food, drinking water, and the human body. To date, there are limited data concerning the toxic action of 2,4,6-TBP and particularly PBP, and no study has been conducted to assess the apoptotic mechanism of action of these substances in human leukocytes. (2) Methods: PBMCs were isolated from leukocyte–platelet buffy coat and treated with tested substances in concentrations ranging from 0.01 to 50 µg/mL for 24 h. The apoptotic mechanism of action of the tested BFRs was assessed by the determination of phosphatidylserine exposure on the PBMCs surface, the evaluation of mitochondrial potential and cytosolic calcium ion levels, and the determination of caspase-8, -9, and -3 activation. Moreover, poly (ADP-ribose) polymerase-1 (PARP-1) cleavage, DNA fragmentation, and chromatin condensation were analyzed. (3) Results: 2,4,6-TBP and, more strongly, PBP induced apoptosis in PBMCs, changing all tested parameters. It was also found that the mitochondrial pathway was mainly involved in the apoptosis of PBMCs exposed to the studied compounds. (4) Conclusions: 2,4,6-TBP and PBP triggered apoptosis in human PBMCs, and some observed changes occurred at 2,4,6-TBP concentrations that were detected in humans occupationally exposed to this substance.

## 1. Introduction

Bromophenolic flame retardants (BFRs), including 2,4,6-tribromophenol (2,4,6-TBP) and pentabromophenol (PBP), are the chemicals widely used in the industry in order to reduce flammability or the burning rate of various materials, including plastics, textiles, and furniture, as well as electric and electronic equipment [1,2].

2,4,6-TBP is the most commonly synthesized bromophenol in the world. Moreover, its production is still increasing because it is widely utilized in the manufacturing of BFRs. In the United States, the production of 2,4,6-TBP was evaluated to be 4500 to <23,000 tons, while in the EU this substance has been recognized as a high-production-volume chemical (HPV) [3,4]. The production volume of PBP is not known, although 55 sources of PBP manufacturing, mainly situated in the USA and China, have been identified [5].

2,4,6-TBP and PBP are mostly utilized as FRs, but they are also used as the intermediates for the manufacturing of allyl ethers [6,7], vinyl-aromatic polymers, and epoxy-phenolic polymers [5]. Moreover, 2,4,6-TBP is used as a wood preservative and fungicide [8], while PBP is utilized as a molluscicide, bactericide, and chemical intermediate for the synthesis of pentabromophenoxy compounds [9,10].

The intensive use of 2,4,6-TBP and PBP has caused the prevalence of these substances in the environment [11,12,13], including biota [14], as well as their presence in indoor air, dust, food, and drinking water [15,16,17,18]. 2,4,6-TBP has been repeatedly identified in humans. For instance, Feng et al. [19] detected 2,4,6-TBP at a mean concentration of 5.57 ± 4.05 µg/L in the urine of the general population of China, whereas Dufour et. al. [20] detected this substance in the range of concentrations from a trace to 1.28 µg/L in the blood of the general population of Belgium. In another study, Gutierrez et al. [21] found high concentrations of 2,4,6-TBP in the urine of Chilean sawmills workers, which were from 1.9 to 12.3 mg/g creatinine (approx. mean 6 mg of 2,4,6-TBP per 1 L of urine). Other studies have shown the presence of 2,4,6-TBP in solid tissues of humans environmentally exposed to this substance. 2,4,6-TBP was found by Smeds and Saukko [22] in adipose tissue (2.16–53.8 μg/kg lipids) collected from people in Finland, whereas Gao et al. [23] detected this compound in concentrations up to 54.3 μg/kg lipids of adipose tissue in inhabitants of New York City, USA. In another study, Leonetti et al. [24] identified 2,4,6-TBP in placental tissue collected from women who delivered term infants (Durham, UK) in significant concentrations ranging from 1.31 to 316 μg/kg lipids. 

It has been proven that BPs are toxic for living organisms, including humans. 2,4,6-TBP and PBP are structurally similar to thyroxine [25], and therefore they are able to bind to nuclear receptors and alter metabolism and the transport of thyroid hormones [26]. Moreover, 2,4,6-TBP has been shown to bind to estrogen and androgen receptors, changing testosterone and estradiol levels in animals [27]. The scientists suggested that the endocrine-disrupting activity and other adverse effects provoked by 2,4,6-TBP and PBP, such as changes in cellular calcium ions or transforming growth factor (TGF-β) signaling pathways, may contribute to cancer development [28,29,30]. In red blood cells, 2,4,6-TBP and PBP increased the level of reactive oxygen species (ROS), oxidized hemoglobin, and induced eryptosis [31,32,33]. Similarly, our previous research revealed that 2,4,6-TBP and PBP induced ROS formation, depleted ATP levels [34], and induced DNA strand breaks and DNA base oxidation in human peripheral blood mononuclear cells (PBMCs) [35]. It is known that ROS induction, the perturbation of energy production, and DNA damage are involved in apoptotic cell death [36].

Apoptosis is a highly regulated process in which the body removes unwanted (old and damaged) cells without an inflammatory state. Nevertheless, it has been shown that various factors, such as toxicants, can accelerate apoptotic cell death, which may lead to the development of different disorders [37].

The assessment of the mechanism of proapoptotic action of toxicants, such as BFRs is crucial for the recognition of biochemical targets that are affected by tested compounds. It must also be taken into consideration that changes in some apoptotic parameters, such as the calcium ion level or transmembrane mitochondrial potential (which usually occur at relatively low toxicant concentrations), are linked to other cellular processes (e.g., cell signaling, the energy state of the cell, etc.), and their disturbance may change cellular function before apoptosis occurs. 

It should also be underlined that halogenated phenols and some BFRs have been shown to exhibit proapoptotic potential in blood cells and other cell types, while their mechanism of action appeared to be complex. Studies have shown that halogenated phenols, such as 2,4,5-trichlorophenol (2,4,5-TCP) and pentachlorophenol (PCP), are capable of inducing apoptosis in human lymphocytes by changing cell membrane permeability and reducing mitochondrial potential and caspase-3 activation [36]. Similarly, Jarosiewicz et al. [37] noticed that 2,4,6-TBP and PBP triggered apoptosis in human erythrocytes through increasing the calcium ion level, caspase-3 activation, and phosphatidylserine translocation on the cell surface. In another study, halogenated brominated diphenyl ethers (PBDEs), such as PBDE-47, PBDE-99, PBDE-209, and particularly 2,2′,4,4′-tetrabromodiphenyl ether (PBDE-47), triggered apoptosis in bronchial epithelial cells by decreasing the transmembrane mitochondrial potential and caspase-3 activation [38]. 

Recently, Dong et al. [39] reported that methylated and acetylated derivatives of natural bromophenols were able to trigger apoptosis in leukemia K562 cells and immortalized human keratocytes (HaCaT cell line).

PBMCs play a key role in the immune system and are permanently exposed to toxicants entering the human body. They are involved in the production of antibodies and responsible for killing virus-infected cells and cancer cells. Moreover, PBMCs participate in the regulation of the immune system response [40]. 

It has been shown that the accelerated apoptosis of PBMCs adversely affects the immune system, causing depressed human immunity [41], which may finally lead to autoimmune diseases, such as type 1 diabetes, allergy, or asthma, as well as cancer development [42,43,44,45].

Some studies have shown that BPs are capable of disturbing the immune system. Xie et al. [46] observed that 2,4,6-TBP caused changes in the production of cytokines, such as tumor necrosis factor (TNF) and interleukins 6 and 10 (IL-6 and IL-10) in RAW264.7 mouse macrophages, as well as the altered polarization of these cells. The authors of this study suggested that the observed effects could have led to a distinct immunomodulatory outcome. In another study, Bowen et al. [47] noticed that PBP changed the phagocytic capability of murine microglial BV-2 cells.

According to our best knowledge, no study has been conducted to assess the effect of BRs, including 2,4,6-TBP and PBP, on apoptosis induction in human leukocytes. In nucleated cells, only Rios et al. [48] and Bowen et al. [47] observed that 2,4,6-TBP and PBP induced apoptosis in neuroblastoma cell cultures (SH-SY5Y) and BV-2 cells, respectively. 

Taking the above into account, we compared the apoptotic potential of 2,4,6-TBP and PBP in human PBMCs and examined the underlying mechanism of action of these substances. In this study, changes in phosphatidylserine exposure (PS) on PBMCs membranes (quantitative determination of apoptosis) and alterations in intracellular calcium ion and transmembrane mitochondrial potential levels, as well as the activation of caspase 8, -9, and -3 were evaluated. Furthermore, the impacts of the examined compounds on poly (ADP-ribose) polymerase-1 (PARP-1) cleavage, DNA fragmentation, and chromatin condensation were studied.

## 2. Results

### 2.1. Quantitative Analysis of Apoptosis

The percent of apoptotic cells in the negative control after 24 h of incubation was approx. 10% (Figure 1), which was due to the significant spontaneous apoptosis of human PBMCs that occurs even when these cells are not exposed to any toxicant. 

2,4,6-TBP and PBP increased the PS exposure of the PBMCs surface, as determined by staining the cells with annexin V-FITC and PI. After 24 h of incubation, PBP at 1 µg/mL caused an increase in the number of apoptotic cells; however, this change was not statistically significant. PBP at higher concentrations of 5 µg/mL and particularly at 25 µg/mL and 50 µg/mL induced substantial (concentration-dependent) statistically significant increases in the number of apoptotic PBMCs. It was also noticed that 2,4,6-TBP induced smaller changes (than PBP) of the tested parameter. 2,4,6-TBP only at 25 µg/mL and 50 µg/mL significantly increased the percentage of apoptotic cells (Figure 1).

### 2.2. Cytosolic Calcium Ion Level

Alterations in intracellular calcium ion level were detected using Fluo-3/AM. This stain is hydrolyzed by membrane esterases to Fluo-3, which after the complexation of calcium ions, shows intense fluorescence. Statistically significant (concentration-dependent) increases in the intracellular calcium ion level were noted in PBMCs treated with PBP at 0.1 µg/mL, 1 µg/mL, and 5 µg/mL after 24 h of incubation. Smaller increases in the cytosolic Ca^2+^ level were noted in PBMCs exposed for 24 h to 2,4,6-TBP, which slightly increased the tested parameter at 1 µg/mL and 5 µg/mL (Figure 2).

### 2.3. Changes in Transmembrane Mitochondrial Potential

The transmembrane mitochondrial potential (Δ*Ψm*) was evaluated using the MitoLite Red CMXRos stain. Alterations in the fluorescence intensity of this probe were associated with changes in the Δ*Ψm* level. It was revealed that 2,4,6-TBP and PBP, in a concentration-dependent manner, reduced Δ*Ψm* in the tested cells after 24 h of incubation. It was shown that PBP at 1 µg/mL, 5 µg/mL, and particularly at 25 µg/mL substantially decreased Δ*Ψm*, whereas 2,4,6-TBP at the same concentrations caused smaller reductions in the tested parameter (Figure 3). 

### 2.4. Caspase-8, -9, and -3 Activation 

Caspase-8 and caspase-9 are initiator caspases of programmed cell death that are involved in the receptor and mitochondrial pathways of apoptosis, respectively. This study showed that both 2,4,6-TBP and PBP more strongly activated caspase-9 than caspase-8 in human PBMCs.

2,4,6-TBP and PBP at 5 µg/mL and 25 µg/mL caused small increases in caspase-8 activation in tested cells after 24 h of incubation (Figure 4A). 

After 24 h of incubation, 2,4,6-TBP and PBP at 5 µg/mL and particularly at 25 µg/mL caused significant increases in caspase-9 activity in PBMCs. It was also noted that 2,4,6-TBP at its highest concentration increased caspase-9 activation more strongly than PBP in the tested cells (Figure 4B). 

Caspase-3 is an executioner caspase of apoptosis. It was found that after 24 h of incubation 2,4,6-TBP and PBP at 5 µg/mL and, more strongly, at 25 µg/mL increased caspase-3 activation in the tested cells, and the effects induced by these substances were similar (Figure 4C).

In all three experiments, preincubation with caspase-8, caspase-9, and caspase-3 inhibitors was performed. The additions of the appropriate caspase inhibitors caused decreases in the enzyme activities to the control values (data not shown).

### 2.5. PARP-1 Cleavage and DNA Fragmentation

PARP-1 is a substrate for caspase-3. It is known that PARP-1 cleavage leads to its inactivation and thus helps the cells undergoing apoptosis. In order to detect PARP-1 degradation, 2,4,6-TBP and PBP were used in the concentration of 25 µg/mL that most efficiently induced caspase-3 activation in the tested cells. 

It was found that after 24 h of incubation 2,4,6-TBP and, more strongly, PBP at 25 µg/mL caused PARP-1 cleavage in human PBMCs (Figure 5A). 

The APO-BrdU TUNEL assay was chosen for labeling DNA strand breaks along with total cellular DNA to determine apoptotic PBMCs. Similarly to the detection of PARP-1 cleavage, the concentration of 25 µg/mL of the tested BRs was selected. 

It was found that after 24 h of incubation 2,4,6-TBP and, more strongly, PBP at 25 µg/mL caused statistically significant increases in the number of TUNEL-positive cells, which proved that the tested BPs caused DNA fragmentation in human PBMCs (Figure 5B). 

### 2.6. *Apoptosis Detection* by Fluorescence *Microscopy*

Figure 6 shows representative photomicrographs of PBMCs stained with Hoechst 33324/PI. Before the cells were stained, they had been treated for 24 h with DMSO at 0.2% (negative control), as well as 2,4,6-TBP or PBP at 5 µg/mL or 50 µg/mL. In control probes, only viable PBMCs were detected. In probes treated with 2,4,6-TBP at 5 µg/mL, mostly viable cells were observed, while in the probes treated with PBP at 5 µg/mL, mainly early apoptotic PBMCs (cells with condensed chromatin) were noticed. The samples treated with 2,4,6-TBP at 50 µg/mL consisted of both viable and late apoptotic cells, while the probes incubated with PBP at 50 µg/mL contained mainly late apoptotic cells.

## 3. Discussion

Apoptosis is a tightly regulated process in which the body eliminates aging or damaged cells without an inflammatory response. Various xenobiotics, such as BFRs can enhance apoptosis, which may lead to the development of infection, heart failure, neurodegenerative disorders, or myocardial ischemia [36]. PBMCs are the primary cells of the immune system, and their excessive removal may contribute to the development of cancer and autoimmune disorders, including diabetes, asthma, or allergy [49].

In this study, the apoptotic mechanism of action of selected BFRs, such as 2,4,6-TBP and PBP, in human PBMCs was studied by analyzing changes in PS translocation and alterations in cytosolic calcium ion and Δ*Ψm* levels, as well as caspase-8, -9, and -3 activation. Moreover, changes in PARP-1 degradation, DNA fragmentation, and chromatin condensation were evaluated. 

It was observed that the tested BPs (5–50 µg/mL after 24 h of incubation) increased the number of apoptotic PBMCs, as assessed by changes in PS translocation, but PBP showed a stronger apoptotic potential than 2,4,6-TBP (Figure 1). Similarly, Jarosiewicz et al. [37] observed that 2,4,6-TBP and, more strongly, PBP (50–100 µg/mL, 48 h of incubation) increased the number of apoptotic red blood cells, as determined by PS exposure on cell surface. In another study, brominated diphenyl ethers (PBDEs), which are used as BFRs, triggered the apoptosis of nucleated cells. Montalbano et al. [38] assessed the toxic effects of PBDE-47, PBDE-99, and PBDE-209 in bronchial epithelial cells, showing that the tested compounds, particularly 2,2′,4,4′-tetrabromodiphenyl ether (PBDE-47) at a low concentration of 1 µM, triggered apoptosis in incubated cells.

The studies have shown that phenol and chlorophenols induce apoptotic alterations in various cell types. Phenol at relatively high concentrations (1.5–2 mM, 24 h of incubation) triggered apoptosis in erythroid progenitor-like K562 cells by increasing the PS exposure on cell surfaces [50], whereas 2,4,5-trichlorophenol (2,4,5-TCP) and pentachlorophenol (PCP) (5–100 µg/mL, 4 h of incubation) induced the apoptosis of human lymphocytes, which was associated with changes in cell membrane permeability that are characteristic for this process [51]. In other studies, Wispriyono et al. [52] observed that PCP at 20 µM (10 h of incubation) induced apoptosis in the Jurkat T-cell line by the stimulation of extracellular signal-regulated kinases and p38 mitogen-activated protein kinases, whereas Chen et al. [53] noticed that 4-chlorophenol (4-CP) 2,4-dichlorophenol (2,4-DCP) and 2,3,6-trichlorophenol (2,3,6-TCP) (0.23–1.09 mM, 6–48 h of incubation) induced concentration- and time-dependent increases in the number of apoptotic mouse L929 fibroblasts, which were associated with the presence of condensed nuclei, segregated nuclei, and apoptotic body formation. 

One of the critical moments in apoptosis is the increase in the cytosolic Ca^2+^ level. The release of Ca^2+^ ions from the cellular compartments of the endoplasmic reticulum into the cytosol can lead to the uptake and accumulation of these ions in the mitochondria, thereby resulting in the reduction in Δ*Ψm* and subsequent apoptotic cell death [54,55,56]. 

Our study showed that 2,4,6-TBP and, more strongly, PBP caused an increase in the intracellular calcium ion level. PBP and 2,4,6-TBP increased this parameter from concentrations of 0.1 µg/mL and 1 µg/mL, respectively (Figure 2). An increase in the Ca^2+^ ion level was observed in the neuroendocrine cell line (PC12) treated with 2,4,6-TBP at 300 µM [26]. In another study, Jarosiewicz et al. [37] noticed that 2,4,6-TBP and PBP at 50 µg/mL (24 h of incubation) increased the cytosolic Ca^2+^ level in human erythrocytes, while Mokra et al. [57] showed that bisphenol A (BPA) and its analogs increased the calcium ion level in the cytosol of PBMCs, which finally led to the apoptosis of treated cells. 

The tested BPs (1–25 µg/mL) caused a decrease in Δ*Ψm* in treated cells, while PBP showed greater changes than 2,4,6-TBP (Figure 3). Bowen et al. [47] observed mitochondria-related effects in the BV-2 microglial cell line exposed to PBP (10–40 µM) for 18 h, whereas Michałowicz and Sicińska [51] reported that 2,4,5-TCP and PCP (1 to 25 µg/mL, 4 h of incubation) depleted Δ*Ψm* in human lymphocytes. In another study, Yan et al. [58] observed that brominated diphenyl ether PBDE-47 in concentrations ranging from 25 to 100 µM (48 h of incubation) induced apoptosis in Jurkat cells (immortalized line of T lymphocytes), which was associated with the overproduction of ROS and a reduction in the Δ*Ψm*. 

An increase in reactive oxygen species (ROS) formation (as observed in our previous study on the prooxidative effects of 2,4,6-TBP and PBP on human PBMCs [34]) and raised cytosolic calcium ion level impair electrolyte transport across the mitochondrial membrane, causing the opening of the mitochondrial permeability transition pores (PTPs), which results in the release of various proapoptotic molecules, such as apoptosis-inducing factor (AIF), cytochrome c, and procaspase-9, into the cytosol [36]. As a consequence, an apoptosome is formed, which leads to caspase-9 activation, and subsequently to the initiation of the intrinsic (mitochondrial) pathway of apoptosis. On the other hand, the external (receptor) pathway of apoptosis may be activated due to the interaction of ligands, including ROS with transmembrane receptors, which results in the activation of caspase-8 [36,56]. Both caspase-8 and caspase-9 may activate executioner caspase-3 [56]. 

In this study, we observed changes in the activation of initiator caspase-8 and -9, as well as executory caspase-3. We found that 2,4,6-TBP and PBP (5–25 µg/mL) slightly increased caspase-8 activity in human PBMCs, while they more strongly (particularly 2,4,6-TBP) caused caspase-9 activation. The obtained results indicate that the mitochondrial pathway was mainly involved in the apoptosis induction of human PBMCs exposed to the tested compounds. The obtained results also showed that the examined BFRs (5–25 µg/mL), to the same extent, increased caspase-3 activation in incubated cells (Figure 4).

Jarosiewicz et al. [37] revealed that 2,4-dibromophenol (2,4-DBP), 2,4,6-TBP, and PBP, from the concentration of 10 µg/mL (48 h of incubation), were capable of increasing caspase-3 activity in red blood cells. Other studies have shown that phenol at high concentrations (1.5–2 mM, 24 h of incubation) increased caspase-3, -8, and -9 activities in erythroid progenitor-like K562 cells [47], whereas 2,4,5-TCP and PCP (5–50 µg/mL, 4 h of incubation) induced caspase-3 activation in human lymphocytes [51]. 

Caspase-3 can cleave PARP-1 into 89 kDa and 24 kDa fragments. The 89 kDa fragments (with PAR polymers) are translocated from the nucleus to the cytoplasm and interact with AIF, which results in shrinkage of the nucleus and apoptosis induction. In contrast, 24 kDa fragments bind to the DNA breaks that are formed during apoptotic cell death [59]. 

This study showed that both 2,4,6-TBP and, more strongly, PBP at 25 µg/mL induced PARP-1 cleavage (Figure 5A). Mokra et al. [57] reported that BPA and its analogues at 5 µg/mL caused PARP-1 cleavage in human PBMCs. In another study, triphenyl phosphate (TPP), which is a phenol derivative used as a FR, caused hepatocyte apoptosis by altering PARP-1 activity [60].

The TUNEL assay is considered to be one of the most-studied and well-known methods for the determination of apoptotic cells, and it is based on the detection of DNA strand breaks in tested cells [61]. 

We noted that PBP and, less strongly, 2,4,6-TBP caused significant increases in the number of TUNEL-positive cells, which confirmed that these substances were able to induce apoptosis in human PBMCs (Figure 5B). The study of Chen et al. [55] showed that 4-CP, 2,4-DCP, and 2,3,4-TCP (0.23–1.09 mM, 6–48 h of incubation) induced apoptosis in mouse L929 fibroblasts, which was associated with DNA fragmentation. 

In order to visualize apoptotic changes, we stained the cells with Hoechst 33342/PI and analyzed them using fluorescence microscopy (Figure 6). The conducted experiment allowed the observation of live, early apoptotic, late apoptotic, and necrotic cells. 

We noticed that PBP at 5 µg/mL caused an increase in the number of early apoptotic cells (cells with clear condensed chromatin), while 2,4,6-TBP at the same concentration almost did not induce apoptosis. Both tested compounds at 50 µg/mL significantly increased the number of apoptotic cells, although PBP induced stronger changes. It was also observed that 2,4,6-TBP and PBP at 50 µg/mL mostly induced the formation of late apoptotic PBMCs (Figure 6). These findings are in agreement with the results of the quantitative determination of apoptosis by means of flow cytometry (Figure 1). Mokra et al. [59] observed that BPA, bisphenol S (BPS), and bisphenol F (BPF) at 100 µg/mL (24 h of incubation) caused chromatin condensation and changes in cell nuclei. Moreover, Chen et al. [53] noticed that mouse L929 fibroblasts incubated with chlorophenols (0.23–1.09 mM, 6–48 h of incubation) showed distinct condensed nuclei, segregated nuclei, and apoptotic bodies formation.

It is worth noting that derivatives of natural BPs also exhibit proapoptotic potential, and this property may be useful in cancer treatment. Dong et al. [39] observed that methylated and acetylated derivatives of natural BPs (5–20 µg/mL, 24 h of incubation) were capable of triggering apoptosis in leukemia K562 cells and immortalized human keratocytes (HaCaT cell line), showing potential anticancer activity. In another study, a natural BP derivative named HPN at a very low concentration (below 1 µM, 12 h of incubation) caused apoptosis in a human liver cancer cell line (HepG2), which was associated with caspase-3 activation, changes in the BAX/Bcl2 ratio, and chromatin condensation [62].

Summing up, the tested BPs were capable of inducing apoptosis in human PBMCs. The mechanism of apoptotic action was assessed, which showed that 2,4,6-TBP and PBP, by changing the cytosolic calcium ion and Δ*Ψm* levels, caused caspase-9 and subsequent caspase-3 activation. The cleavage of PARP-1 by caspase-3 resulted in DNA fragmentation and chromatin condensation in the tested cells. The observed activation of caspase-8 seems to have minor importance for the apoptotic changes observed in PBMCs treated with the tested BFRs.

It must be underlined that changes in the intracellular calcium ion level, Δ*Ψm*, and caspase-3, -8, and -9 activation were observed in human PBMCs treated with 2,4,6-TBP in the concentrations that were determined in humans occupationally exposed to this substance [21]. As no data, according to our best knowledge, on the presence of PBP in the human body exist, it is difficult to conclude whether this substance, in the concentrations used in this study, may cause apoptotic changes in humans. Nevertheless, it must be noted that PBP caused stronger apoptotic alterations than 2,4,6-TBP, increasing the intracellular Ca^2+^ level and the number of apoptotic cells, even at concentrations of 0.1 µg/mL and 5 µg/mL, respectively. 

## 4. Conclusions

In conclusion, (1) 2,4,6-TBP and, more strongly, PBP exhibited apoptotic potential in human PBMCs. (2) The studied BPs triggered apoptosis by inducing PS exposure on cell surface; they also elevated the cytosolic calcium ion level, depleted the Δ*Ψm*, activated caspase-8, -9, and -3 and caused PARP-1 cleavage, DNA fragmentation, and chromatin condensation. (3) 2,4,6-TBP and PBP induced apoptosis, mainly by the involvement of the mitochondrial pathway, while the receptor pathway was of minor importance for apoptotic cell death activation. (4) Apoptotic changes in PBMCs were not observed at BFRs concentrations determined in humans environmentally exposed to these substances; however, changes in the intracellular calcium ion level and Δ*Ψm*, as well as alterations in caspases activities occurred in studied cells treated with 2,4,6-TBP in the concentrations that were determined in humans occupationally exposed to this compound. 

## 5. Materials and Methods

### 5.1. Chemicals

2,4,6-Tribromophenol (2,4,6-TBP) and pentabromophenol (PBP) of 99% purity were purchased from LGC Standards (Teddington, UK). HBSS solution, valinomycin, pluronic F-127, propidium iodide, Hoechst 33342, and a Caspase-3 Fluorimetric Assay Kit were bought from Sigma-Aldrich (St. Louis, MO, USA). FITC Annexin V Apoptosis Detection Kit and Perm/Wash Buffer were bought from Becton Dickinson (Franklin Lakes, NJ, USA). MitoLite Red CMXRos was obtained from AAT Bioquest (Sunnyvale, CA, USA). Fluo-3/AM was obtained from PromoCell (Heidelberg, Germany). A Caspase-8 Fluorimetric Assay Kit, as well as a caspase-9 chromogenic substrate and caspase-9 inhibitor were purchased from BioVision (San Francisco, CA, USA). An APO-BrdU TUNEL Assay Kit and PARP-1 (cleaved Asp214) monoclonal antibody (HLNC4) were obtained from Thermo-Fisher (Waltham, MA, USA). Lymphocyte separation medium (LSM) (1.077 g/cm^3^) and RPMI medium with L-glutamine were purchased from Cytogen (Seoul, South Korea). Camptothecin was obtained from Pol-Aura, Poland, while ionomycin was purchased from Biokom (Janki, Poland). Other chemicals were of analytical grade and were obtained from POCH, Poland, and Roth, Germany.

### 5.2. Methods

#### 5.2.1. Cell Isolation and Treatment

PBMCs were isolated from the leukocyte–platelet buffy coat that was achieved from the whole blood in the Blood Bank in Lodz, Poland. Blood was collected from healthy non-smoking volunteers (aged 18–45) who did not show any signs of symptoms of infectious disease. The method of isolation of PBMCs was described in detail in the study conducted by Włuka et. al. [31]. This research was approved by the Bioethical Commission of Scientific Research at the University of Lodz, no. 1/KBBN-UŁ/II/2017. 

The examined compounds were dissolved in DMSO that had a final concentration of 0.2% in the untreated samples (negative control), as well as in the samples treated with 2,4,6-TBP or PBP. The DMSO concentration used in this study was not toxic for PBMCs, as assessed by all analyzed parameters. 

The cells were treated with the examined substances in concentrations ranging from 0.01 to 50 µg/mL (depending on the method used) for 24 h at 37 °C in 5% CO_2_ atmosphere in total darkness. The final density of PBMCs used in the experiments (after BFR solution addition) was 4 × 10^6^ cells/mL for the caspase analysis and 1 × 10^6^ cells/mL for the determination of the other examined parameters. The viability of PBMCs in the negative control samples was over 95%. 

The lowest concentration of examined BFRs (0.01 µg/mL) used in this research (the analysis of the cytosolic calcium ion level and the transmembrane mitochondrial potential) corresponded to the mean 2,4,6-TBP level detected in the general human population of China, as reported by Feng et al. [19].

#### 5.2.2. Quantitative Determination of Apoptosis (Annexin V-FITC/PI Staining)

This method is based on the ability of annexin V labeled with fluorescein isothiocyanate (FITC) to bind to PS, which is transferred on the outer monolayer of the cellular membrane of apoptotic cells. PI is used to determine necrotic cells, as it penetrates damaged membranes and binds with DNA. This experiment was performed according to the procedure given by the manufacturer for the Annexin V-FITC apoptosis detection kit. 

PBMCs were treated with 2,4,6-TBP or PBP in a range of concentrations from 1 to 50 µg/mL and incubated for 24 h at 37 °C in total darkness. After incubation, the cells were centrifuged (300× *g*) for 5 min at 4 °C and suspended in RPMI medium. Then, PBMCs were stained with the mixture of Annexin V-FITC and PI (1 µM each) dissolved in Annexin V-binding buffer and incubated for 20 min at room temperature in total darkness. In the cells, apoptosis was triggered with camptothecin at 10 μM (positive control). The probes were determined using flow cytometry (LSR II, Becton Dickinson) (excitation/emission maxima: 488/525 nm for annexin V and 530/620 nm for PI, respectively). The FMC gate on the PBMCs was established, and the data were recorded for a total of 10,000 cells per sample. 

#### 5.2.3. Cytosolic Calcium Ion Level

An increase in the intracellular Ca^2+^ level has been recognized as one of the primary features of apoptosis. This parameter was determined by means of Fluo-3/AM staining. Fluo-3/AM shows negligible fluorescence; however, after hydrolysis by membrane esterases (fluo-3 formation) and complexation with calcium ions, it shows about a 100-fold increase in green fluorescence intensity. 

The cells were treated with 2,4,6-TBP or PBP in a range of concentrations from 0.01 to 5 µg/mL and incubated for 24 h at 37 °C in total darkness. Then, PBMCs were centrifuged (300× *g*) for 5 min at 4 °C, resuspended in Fluo-3 AM solution (1 µM), and incubated for 20 min at 37 °C in total darkness. In the next step, HBSS containing 1% BSA was added to the cells that were incubated for 40 min at 37 °C in total darkness. PBMCs were washed twice using HEPES buffer and centrifuged (300× *g*) for 5 min at 4 °C. In the final step, the cells were resuspended in HEPES buffer and incubated for 10 min at 37 °C in total darkness. The positive control consisted of PBMCs exposed to ionomycin at 1 µM (calcium ionophore). The probes were analyzed by flow cytometry (LSR II, Becton Dickinson) (excitation/emission maxima: 488/525 nm for fluo-3). The FMC gate on PBMCs was established, and the data were recorded for a total of 10,000 cells per sample.

#### 5.2.4. Mitochondrial Transmembrane Potential (Δ*Ψm*)

A depletion of Δ*Ψm* is considered to be a feature of early apoptosis. Δ*Ψm* was detected based on alterations in the intensity of the fluorescence of MitoLite Red CMXRos (excitation/emission maxima: 579/599 nm). This stain is a cationic dye that is capable of entering living cells and bioaccumulating in mitochondria, depending on the Δ*Ψm* level. The stain is able to remain in mitochondria because it consists of thiol-reactive chloromethyl moieties. 

PBMCs were treated with 2,4,6-TBP or PBP in a range of concentrations from 0.01 to 25 µg/mL for 24 h at 37 °C in total darkness. Nigericin and valinomycin (1 µM), which are capable of increasing and decreasing Δ*Ψm*, respectively, were used as positive controls. After the probes had been incubated, they were centrifuged (300× *g*) for 5 min at 4 °C. The supernatant was discarded, and the cells were resuspended in PBS solution. PBMCs were stained with MitoLite CMXRos at 1 µM and then incubated for 20 min at 37 °C in total darkness. The probes were determined in 96-well plates using a microplate reader (Cary Eclipse, Varian).

#### 5.2.5. Caspase-3, -8, and -9 Activation

Apoptosis is regulated directly and indirectly by caspases. A fluorimetric analysis of caspase-3 and caspase-8 activity was conducted according to the manufacturers′ protocols. The methods of caspase-3 and caspase-8 determination are based on the hydrolysis (by these enzymes) of peptide substrates, such as acetyl-Asp-Glu-Val-Asp-7-amino-4-methylcoumarin (Ac-DEVD-AMC) and acetyl-Ile-GluThr-Asp-7-amino-4-methylcoumarin (Ac-IETD-AMC), respectively. The hydrolysis of the substrates leads to the release of the fluorescent 7-amino-4-methylcoumarin (AMC) (excitation/emission maxima: 360/460 nm). The colorimetric evaluation of caspase-9 activity was due to the hydrolysis (by this enzyme) of the substrate acetyl-Leu-Glu-His-Asp-p-nitroaniline (Ac-LEHD-pNA), which caused a release of p-nitroaniline (pNA) (absorption at 405 nm). In all experiments, positive controls were used that consisted of PBMCs suspensions incubated with camptothecin (10 µM). Preincubation with caspase-3, caspase-8, and caspase-9 inhibitors was also conducted for all experiments. The detection of caspase-3 and caspase-8 activities was performed using a fluorescent microplate reader (Fluoroskan Ascent FL, Labsystem), while the determination of caspase-9 activity was performed using an absorbance microplate reader (BioTek ELx808, Bio-Tek) (Winooski, VT, USA).

#### 5.2.6. PARP-1 Cleavage

PARP-1 is the enzyme present in the nucleus that is involved in DNA repair and is implicated in other essential cellular processes. When caspase-3 is activated during apoptosis, it cleaves PARP-1 (between Asp214 and Gly215), leading to the formation of two fragments of 85 kDa and 25 kDa. The addition of an HLNC4 antibody (conjugated with Alexa Fluor 488) allows the specific detection of the 85 kDa PARP-1 fragment. 

The cells were treated with 2,4,6-TBP or PBP at 25 µg/mL and incubated for 24 h at 37 °C in total darkness. Then, PBMCs were washed and suspended in 1% paraformaldehyde (dissolved in PBS solution). Finally, an HLNC4 antibody conjugated with Alexa Fluor 488 was added to the samples that were incubated for 30 min at 37 °C in total darkness. In the cells, apoptosis was induced by camptothecin at 10 μM (positive control). A cytometric analysis of the samples was performed (LSR II, Becton Dickinson) at excitation/emission maxima of 494/519 nm for Alexa Fluor 488. The FMC gate on PBMCs was established, and the data were recorded for a total of 10,000 cells per sample.

#### 5.2.7. APO-BrdU TUNEL Assay

The TUNEL method allows the determination of apoptosis by labeling the 3′-OH ends of single- and double-stranded DNA fragments with labeled brominated deoxyuridine triphosphate nucleotides (Br-dUTP). The reaction is catalyzed by terminal deoxynucleotidyl transferase (TdT). 

The cells were treated with 2,4,6-TBP or PBP at 25 µg/mL and incubated for 24 h at 37 °C in total darkness. Then, PBMCs were fixed in paraformaldehyde (1%). In the next step, the samples were incubated in a DNA labeling solution containing BrdUTP and TdT for 1 h at 37 °C in total darkness. Finally, an anti-BrdUTP antibody was added to the probes that were incubated for 30 min at room temperature in total darkness. In the cells, apoptosis was induced by camptothecin at 10 μM (positive control). The samples were determined by flow cytometry (LSR II, Becton Dickinson) at excitation/emission maxima of 494/519 nm for Alexa Fluor 488. The FMC gate on PBMCs was established, and the data were recorded for a total of 10,000 cells per sample.

#### 5.2.8. Hoechst 33342/PI Staining

Apoptotic changes (particularly chromatin condensation) were observed using fluorescence microscopy. PBMCs were incubated with the tested BFRs and then stained with Hoechst 33342 and PI. Based on morphological features, PBMCs were described as viable (blue fluorescence), early apoptotic (intense bright-blue fluorescence), late apoptotic (blue-violet fluorescence), and necrotic (red fluorescence) [63]. PBMCs were treated with 2,4,6-TBP or PBP at 5 µg/mL or 50 µg/mL and incubated for 24 h at 37 °C in total darkness. After incubation, the cells were centrifuged (200× *g*) for 3 min at 4 °C, then the supernatant was discarded. PBMCs were resuspended in a PBS solution (0.5 mL), and the mixture of 1 µL of Hoechst 33342 and 1 µL of PI (1 mg/mL each) was added to the probes. Finally, the cells were incubated for 1 min at 37 °C in total darkness and analyzed using a fluorescence microscope (Olympus IX70, Japan) at 400× magnification. 

#### 5.2.9. Statistical Analysis

Data are shown as average values with standard deviations. The ANOVA (one-way analysis of variance) test and Tukey’s post hoc test or Welch’s test were employed to evaluate the statistical significance between the examined probes [64]. Statistical significance was considered to be *p* < 0.05. All statistical evaluations were conducted using STATISTICA 13 software (StatSoft, Inc, Tulusa, OK, USA). The tests were conducted on blood from four donors, while for each individual experiment (one blood donor), an experimental point was the mean value of 2–3 replications.

In the statistical analyses, the results (Figure 2, Figure 3, Figure 4 and Figure 5) were recounted as the % of control. The raw control data did not differ in any significant way between each another; however, the standardization of control values allowed for a better assessment of the impact of the tested compounds on the examined parameters. According to the literature data (Watała, 2002), the results recalculated as the % of control are considered to be relative values, which are always recognized in statistical analysis as unpaired. The statistical analysis of this type of data included sequentially testing for the normality of the distribution (Shapiro–Wilk test) and variance (Brown–Forsythe test). Finally, due to the normal distribution and homogeneity of variance, our data were analyzed using an ANOVA and post hoc test (Tukey’s test).

## Figures and Tables

**Figure 1 molecules-27-05056-f001:**
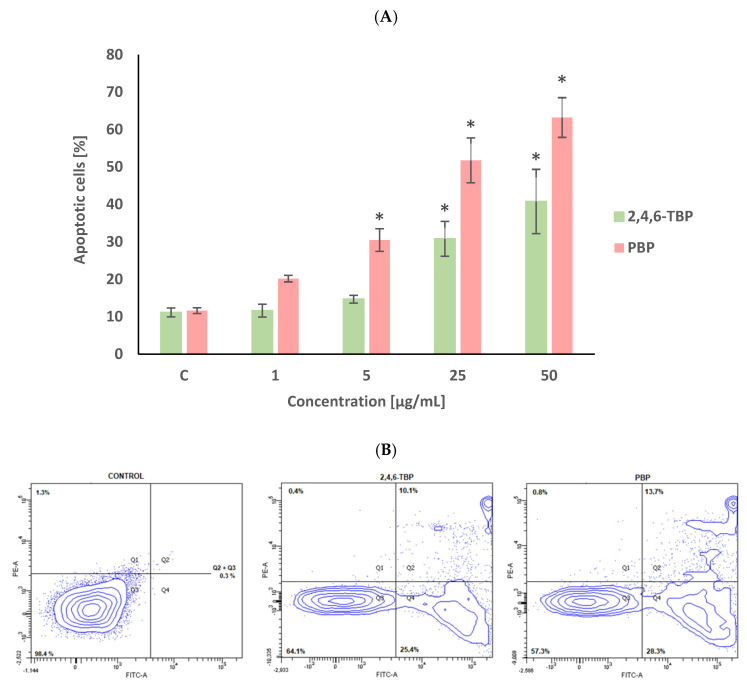
Apoptotic changes in human PBMCs incubated with 2,4,6-TBP and PBP in concentrations ranging from 1 to 50 µg/mL for 24 h (**A**). The cells were stained with Annexin V-FITC and propidium iodide. Exemplary dot plot showing apoptotic changes in human PBMCs unexposed (negative control), as well as exposed to 2,4,6-TBP at 50 μg/mL and PBP at 25 μg/mL for 24 h, Q1—necrotic cells, Q2 + Q3—apoptotic cells, Q4—live cells (**B**). (*) Means ± SDs were calculated from four individual experiments (four blood donors). Results were statistically different from negative control at * *p* < 0.05. Statistical analysis was conducted using one-way ANOVA and a posteriori Tukey test.

**Figure 2 molecules-27-05056-f002:**
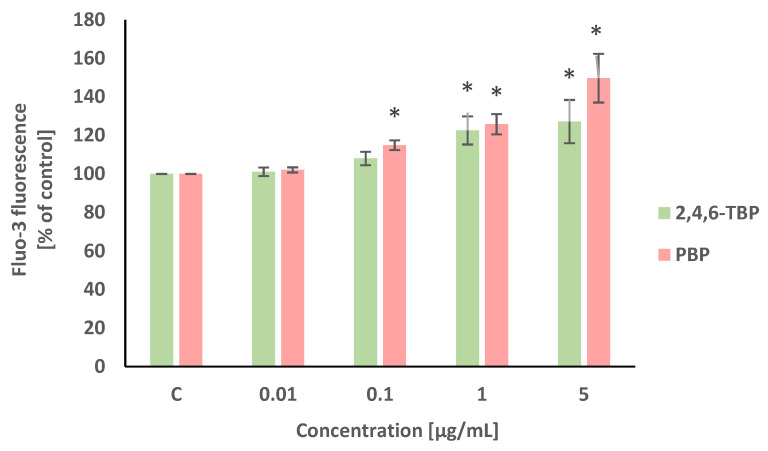
Changes in intracellular calcium ion levels in human PBMCs incubated with 2,4,6-TBP and PBP in concentrations ranging from 0.01 to 5 µg/mL for 24 h. Means ± SDs were calculated from four individual experiments (four blood donors). Results were statistically different from negative control at * *p* < 0.05. Statistical analysis was conducted using one-way ANOVA and a posteriori Tukey test.

**Figure 3 molecules-27-05056-f003:**
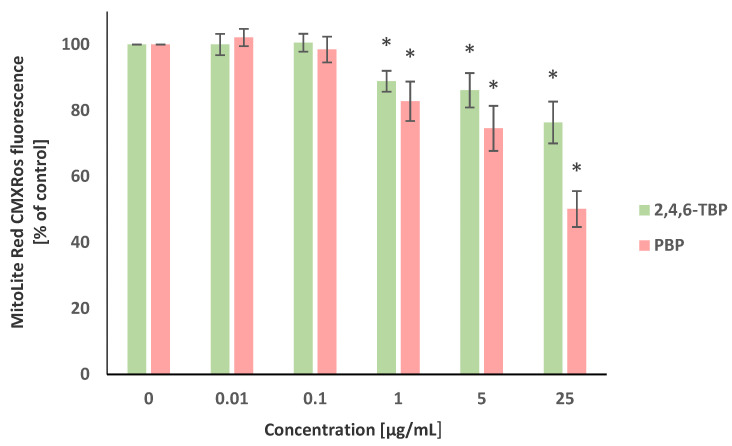
Changes in transmembrane mitochondrial potential (Δ*Ψm*) of PBMCs incubated with 2,4,6-TBP and PBP in concentrations ranging from 0.01 to 25 µg/mL for 24 h. Means ± SDs were calculated from four individual experiments (four blood donors). Results were statistically different from negative control at * *p* < 0.05. Statistical analysis was conducted using one-way ANOVA and a posteriori Tukey test.

**Figure 4 molecules-27-05056-f004:**
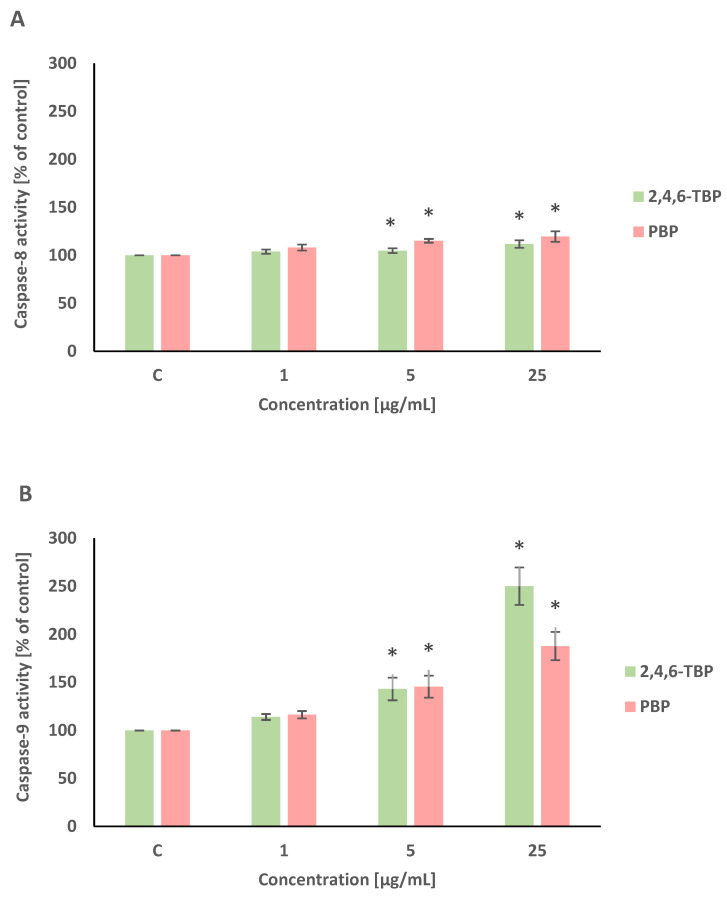
Changes in the activity of caspase-8 (**A**), -9 (**B**), and -3 (**C**) in human PBMCs incubated with 2,4,6-TBP and PBP in concentrations ranging from 1 to 25 µg/mL for 24 h. Means ± SDs were calculated from four individual experiments (four blood donors). Results were statistically different from negative control at * *p* < 0.05. Statistical analysis was conducted using one-way ANOVA and a posteriori Tukey test.

**Figure 5 molecules-27-05056-f005:**
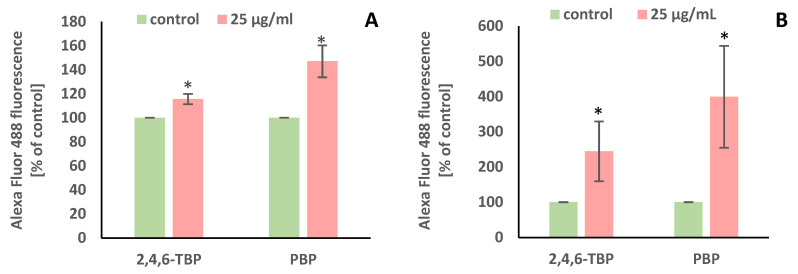
Changes in PARP-1 cleavage (**A**) and DNA fragmentation (**B**) in human PBMCs incubated with 2,4,6-TBP and PBP in the concentration of 25 µg/mL for 24 h. Means ± SDs were calculated from four individual experiments (four blood donors). Results were statistically different from negative control at * *p* < 0.05. Statistical analysis was conducted using Welch’s test.

**Figure 6 molecules-27-05056-f006:**
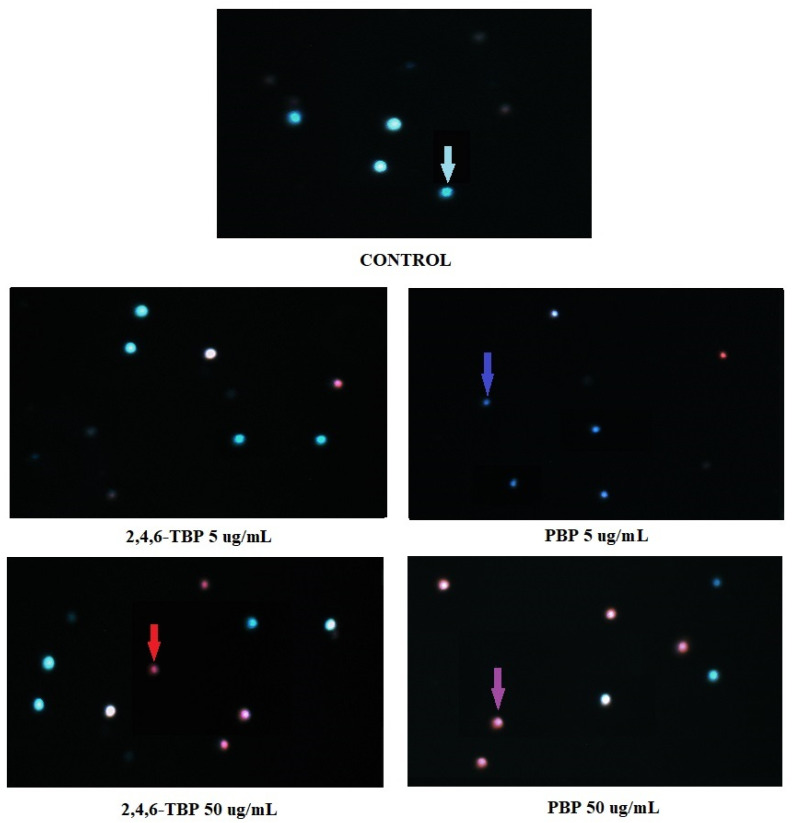
Apoptotic changes in PBMCs incubated with DMSO (control) as well as 2,4,6-TBP and PBP at 5 µg/mL and 50 µg/mL. The cells were stained with Hoechst 33342 and PI. Viable cells (blue fluorescence), early apoptotic cells (intense bright-blue fluorescence), late apoptotic cells (blue/violet fluorescence), and necrotic cells (red fluorescence).

## Data Availability

The raw data supporting the conclusions of this paper are deposited in the Department of Biophysics of Environmental Pollution, University of Lodz, and will be made available by the authors without undue reservation.

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
