# Peer review of "Apoptosis-Inducing Potential of Selected Bromophenolic Flame Retardants 2,4,6-Tribromophenol and Pentabromophenol in Human Peripheral Blood Mononuclear Cells"

_molecules, 2022, doi:10.3390/molecules27165056_

Round 1

Reviewer 1 Report

Baranska et al. studied the effect of 2,4,6-tribromophenol (TBT) and pentabromophenol in human peripheral blood mononuclear cells. The study is well planned and executed. On the other hand, interpretation of the results is misleading. As the authors found, the compounds induced apoptosis at 5 μg/mL concentration and above, which exceeds the 22-360 pg/g ww found in human blood and tissues (Koch and Sures, Environmental Pollution 233 (2018), 706-713) at least by four orders of magnitude. The authors are advised to establish the relevance of their study in the Introduction section more firmly. Also, they are requested to downplay their conclusion of apoptosis-inducing potential of the bromophenol compounds at concentrations found in Human.

Specific point:

1. In the report by Gutierrez et al. (reference 21), 1.9 to 12.3 mg/g creatinine of TBP was detected in the urine of Chilean sawmills workers. As the urine contains 2.5 to 8.2 mM creatinine, the mean TBT content of the urine was about 6 mg/L, which is 1000 times lower than the lowest effective TBT concentration the authors used. Furthermore, as TBT is excreted mainly in the urine (see e.g. Koch and Sures 2018), urine content likely exceeded blood content. Regardless, reference 21 is contrary rather than supporting for the authors' conclusion. The conclusion should be modified accordingly.

Author Response

Baranska et al. studied the effect of 2,4,6-tribromophenol (TBT) and pentabromophenol in human peripheral blood mononuclear cells. The study is well planned and executed. On the other hand, interpretation of the results is misleading. As the authors found, the compounds induced apoptosis at 5 μg/mL concentration and above, which exceeds the 22-360 pg/g ww found in human blood and tissues (Koch and Sures, Environmental Pollution 233 (2018), 706-713) at least by four orders of magnitude. The authors are advised to establish the relevance of their study in the Introduction section more firmly. Also, they are requested to downplay their conclusion of apoptosis-inducing potential of the bromophenol compounds at concentrations found in Human.

Specific point:

In the report by Gutierrez et al. (reference 21), 1.9 to 12.3 mg/g creatinine of TBP was detected in the urine of Chilean sawmills workers. As the urine contains 2.5 to 8.2 mM creatinine, the mean TBT content of the urine was about 6 mg/L, which is 1000 times lower than the lowest effective TBT concentration the authors used. Furthermore, as TBT is excreted mainly in the urine (see e.g. Koch and Sures 2018), urine content likely exceeded blood content. Regardless, reference 21 is contrary rather than supporting for the authors' conclusion. The conclusion should be modified accordingly.

Dear Reviewer

Thank you very much for favorable evaluation of our manuscript and your valuable comments:

We agree with reviewer that changes in apoptotic parameters were observed in PBMCs at much higher concentration than those, which have been found in humans environmentally exposed to 2,4,6-tribromophenol. In the entire manuscript there is no suggestion that observed apoptotic changes, including an increased number of apoptotic cells were determined at environmentally relevant concentrations of 2,4,6-TBP. It is worth noting that we analyzed much lower concentrations of studied BFRs, including 2,4,6-TBP (Figure 2 and 3), which were similar to those detected in Chinese general population (mean 5.57±4.05 µg/L) (Feng et al. 2016), but as mentioned above no statistically significant changes were observed at this level.

Feng, C.; Xu, Q.; Jin, Y.; Lin, Y.; Qui, X.; Lu, D.; Wang, G. Determination of urinary bromophenols (BrPs) as potential biomarkers for human exposure to polybrominated diphenyl ethers (PBDEs) using gas chromatography-tandem mass spectrometry (GC–MS/MS). J. Chromat. B. 2016, 1022, 70-74.

In the line of reviewer comment, we added the following sentence to ‘Introduction’ to organize the data on environmental and occupational exposure of humans to 2,4,6-TBP, i.e.

‘Other studies have shown the presence of 2,4,6-TBP in solid tissues of humans environmentally exposed to this substance.’

In the case of environmental exposure, as reviewer pointed out, the scientific report of Gutierrez et al. (2005) showed that workers can be exposed to huge amounts of 2,4,6-TBP, which urinary level was determined to be 1000 times higher than in the case of environmental exposure.

We agree with reviewer that calculated mean 2,4,6-TBP concentration should be rather 6 mg/L than 10 mg/L (this corrected value was included in Introduction).

However, we cannot agree with reviewer statement that 2,4,6-TBP concentration of 6 mg/L is ‘1000 times lower than the lowest effective TBT concentration the authors used’.

The 2,4,6-TBP concentration of 6 mg/L equals 6 µg/mL, which is higher than those, which increased calcium ion level (1 µg/mL), decreased transmembrane mitochondrial potential (1 µg/mL), as well as increased caspase-3, -8 and -9 activities (even, if we consider that urinary 2,4,6-TBP concentration is higher than its content in blood); therefore in our opinion the information contained in the last point of conclusion of our manuscript is not incorrect.

Nevertheless, we modified this point of conclusion to provide more clear information for the readers:

‘(4) Apoptotic changes in PBMCs were not observed at BFRs concentrations determined in humans environmentally exposed to these substances; however changes in intracellular calcium ion level and transmembrane mitochondrial potential, as well as alterations in caspases activities occurred in studied cells treated with 2,4,6-TBP in the concentrations that were determined in humans occupationally exposed to this compound.’

Reviewer 2 Report

A revised version is significantly improved, it could be accepted for publication. I only recommend to include into the text of the paper authors' response (in a few sentences) to my remarks No.4 (statistical analysis) and No. 8 (high baseline levels of apoptotic cells in control cultures)

Author Response

A revised version is significantly improved, it could be accepted for publication. I only recommend to include into the text of the paper authors' response (in a few sentences) to my remarks No.4 (statistical analysis) and No. 8 (high baseline levels of apoptotic cells in control cultures)

Dear Reviewer

Thank you very much for favorable evaluation of our manuscript and your valuable comments:

According to reviewer suggestion the information contained in the responses referringto statistical analysis and baseline level of apoptotic cells in control values was added into the manuscript (methods and results sections, respectively).
